# Implicit Bias of Mirror Flow on Separable Data

Scott Pesme
EPFL

Radu-Alexandru Dragomir
Télécom Paris

Nicolas Flammarion
EPFL

## Abstract

We examine the continuous-time counterpart of mirror descent, namely mirror flow, on classification problems which are linearly separable. Such problems are minimised 'at infinity' and have many possible solutions; we study which solution is preferred by the algorithm depending on the mirror potential. For exponential tailed losses and under mild assumptions on the potential, we show that the iterates converge in direction towards a $\phi_\infty$-maximum margin classifier. The function $\phi_\infty$ is the *horizon function* of the mirror potential and characterises its shape 'at infinity'. When the potential is separable, a simple formula allows to compute this function. We analyse several examples of potentials and provide numerical experiments highlighting our results.

## 1 Introduction

Heavily over-parametrised yet barely regularised neural networks can easily perfectly fit a noisy training set while still performing very well on unseen data [Zhang et al., 2017]. This statistical phenomenon is surprising since it is known that there exists interpolating solutions which have terrible generalisation performances [Liu et al., 2020]. To understand this benign overfitting, it is essential to take into account the training algorithm. If overfitting is indeed harmless, it must be because the optimisation process has steered us towards a solution with favorable generalisation properties.

From this simple observation, a major line of work studying the *implicit regularisation* of gradient methods has emerged. These results show that the recovered solution enjoys some type of low norm property in the infinite space of zero-loss solutions. Gradient descent (and its variations) has therefore been analysed in various settings, the simplest and most emblematic being that of gradient descent for least-squares regression: it converges towards the solution which has the lowest $\ell_2$ distance from the initialisation [Lemaire, 1996]. In the classification setting with linearly separable data, iterates of gradient methods must diverge to infinity to minimise the loss. Therefore, the directional convergence of the iterates is considered and Soudry et al. [2018] show that gradient descent selects the $\ell_2$-max-margin solution amongst all classifiers.

Going beyond linear settings, it has been observed that **an underlying mirror-descent structure very recurrently emerges** when analysing gradient descent in a range of non-linear parametrisations [Woodworth et al., 2020, Azulay et al., 2021]. Providing convergence and implicit regularisation results for mirror descent has therefore gained significant importance.

In this context, for linear regression, Gunasekar et al. [2018] show that the iterates converge to the solution that has minimal Bregman distance to the initial point. Turning towards the classification setting, an apparent gap emerges as there is still no clear understanding of what happens: Can directional convergence be characterised in terms of a max-margin problem? If so, what is the associated norm? Quite surprisingly, this question remains largely unanswered, as it is only understood for $L$-homogeneous potentials [Sun et al., 2023]. Our paper bridges this gap by formally characterising the implicit bias of mirror descent for separable classification problems.

38th Conference on Neural Information Processing Systems (NeurIPS 2024).

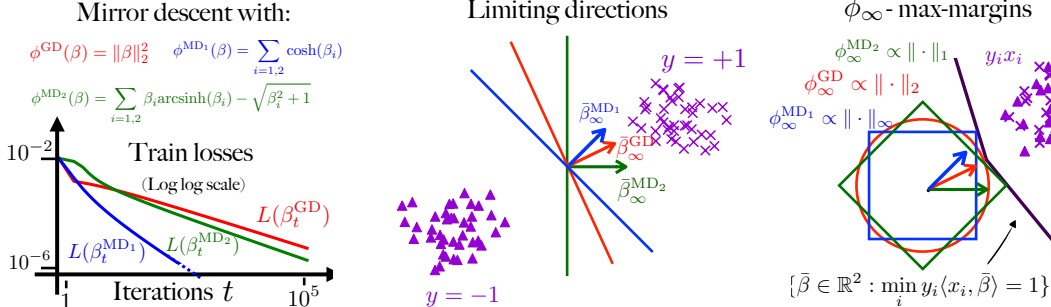

Figure 1: Mirror descent is performed using 3 different potentials on the same toy 2d dataset. *Left:* the losses converge to zero. *Center:* the iterates converge in direction towards 3 different vectors $\bar{\beta}_\infty$, the 3 lines passing through the origin correspond to the associated separating hyperplanes. *Right:* the limit directions are each proportional to $\arg\min \phi_\infty(\bar{\beta})$ under the constraint $\min_i y_i \langle x_i, \bar{\beta}\rangle \geq 1$ for their respective $\phi_\infty$'s, as predicted by our theory (Theorem 1). The full trajectories are plotted Figure 4 and we refer to Section 5 for more details.

## 1.1 Informal statement of the main result

For a separable dataset $(x_i, y_i)_{i \in [n]}$, we study the mirror flow $\mathrm{d}\nabla\phi(\beta_t) = -\nabla L(\beta_t)\mathrm{d}t$ with potential $\phi : \mathbb{R}^d \to \mathbb{R}$ and an exponential tailed classification loss $L$. We prove that $\beta_t$ converges in direction towards the solution of the $\phi_\infty$-maximum margin solution where the (asymmetric) norm $\phi_\infty$ captures the shape of the potential $\phi$ 'at infinity' (see Figure 2 for an intuitive illustration).

**Theorem 1** (Main result, Informal). *There exists a **horizon function** $\phi_\infty$ such that for any separable dataset, the normalised mirror flow iterates $\bar{\beta}_t := \beta_t/\|\beta_t\|$ converge and satisfy:*

$$\lim_{t \to \infty} \bar{\beta}_t \quad \text{is proportional to} \quad \underset{\min_i y_i\langle x_i, \bar{\beta}\rangle \geq 1}{\arg\min} \phi_\infty(\bar{\beta}).$$

Our result holds for a large class of potentials $\phi$ and recovers previous results obtained for $\phi = \|\cdot\|_p^p$ [Sun et al., 2022] and for $L$-homogeneous potentials [Sun et al., 2023]. For general potentials, showing convergence towards a maximum margin classifier is much harder because, in stark contrast with homogeneous potentials, $\phi$'s geometry changes as the iterates diverge. To capture the behaviour of $\phi$ at infinity, we geometrically construct its horizon function $\phi_\infty$. By considering $\phi$'s successive level sets (and re-normalising them to prevent blow up), we show that under mild assumptions, these sets asymptotically converge towards a limiting horizon set $S_\infty$. The horizon function $\phi_\infty$ is then simply the asymmetric norm which has $S_\infty$ as its unit ball (see Figure 3 for an illustration). In addition, when the function $\phi$ is 'separable' and can be written $\phi(\beta) = \sum_i \varphi(\beta_i)$ for a real valued function $\varphi$, then a very simple and explicit formula enables to calculate $\phi_\infty$ (Theorem 3).

The paper is organised as follows. The classification setting as well as the assumptions on the loss and the potential are provided in Section 2. The proof sketch and an intuitive construction of the horizon function are given in Section 3. In Section 4, we state the formal definition and results. Simple examples of horizon potentials and numerical experiments supporting our claims are finally given in Section 5.

## 1.2 Relevance of mirror descent and related work

We first outline the motivations for understanding the implicit regularization of mirror descent and discuss related works that contextualize our contribution within the machine learning context.

**Relevance of studying mirror descent in the context of machine learning.** Though mirror descent is not *per se* an algorithm used by machine learning practitioners, it proves to be a very useful tool for theoreticians in the field. Indeed, when analysing gradient descent (and its stochastic and accelerated

variants) on neural-network architectures, an underlying mirror-descent structure very recurrently emerges. Then, results for mirror descent enable to prove convergence as well as characterise the implicit bias of gradient descent for these architectures. Diagonal linear networks, which are ideal proxy models for gaining insights on complex deep-learning phenomenons, is the most notable example of such an architecture. The hyperbolic entropy potential naturally appears and enables to prove countless results: implicit bias of gradient descent in regression [Woodworth et al., 2020, Vaskevicius et al., 2019] and in classification [Moroshko et al., 2020], effect of stochasticity [Pesme et al., 2021] and momentum [Papazov et al., 2024], convergence of gradient descent and effect of the step-size [Even et al., 2023], saddle-to-saddle dynamics [Pesme and Flammarion, 2023]. Unveiling an underlying mirror-like structure goes beyond these simple networks as they also appear in: matrix factorisation with commuting observations [Gunasekar et al., 2017, Wu and Rebeschini, 2021], fully connected linear networks [Azulay et al., 2021, Varre et al., 2023] and 2-layer ReLU networks [Chizat and Bach, 2020]. Building on these examples, Li et al. [2022] investigate the formal conditions that ensure the existence of a mirror flow reformulation for general parametrisations, extending previous results by Amid and Warmuth [2020a,b].

**Gradient descent in classification.** Numerous works have studied gradient descent in the classification setting. For linear parametrisations, separable data and exponentially tailed losses, Soudry et al. [2018] prove that GD converges in direction towards the $\ell_2$-maximum margin classifier and provides convergence rates. A very fine description of this divergence trajectory is conducted by Ji and Telgarsky [2018] and a different primal-dual analysis leading to tighter rates is given by Ji and Telgarsky [2021]. Similar results are proven for stochastic gradient descent by Nacson et al. [2019c]. In the case of general loss tails, Ji et al. [2020] prove that gradient descent asymptotically follows the $\ell_2$-norm regularisation path. A whole 'astral theory' is developed by Dudík et al. [2022] who provide a framework which enables to handle 'minimisation at infinity'. Beyond the linear case, Lyu and Li [2020] proves for homogeneous neural networks that any directional limit point of gradient descent is along a KKT point of the $\ell_2$-max margin problem. A weaker version of this result was previously obtained by Nacson et al. [2019a]. Furthermore, convergence results for linear networks are provided by Yun et al. [2021]. Finally, for 2-layer networks in the infinite width limit, assuming directional convergence, Chizat and Bach [2020] proves that the limit can be characterised as a max-margin classifier in a certain space of functions.

## 1.3 Notations

We provide here a few notations which will be useful throughout the paper. We let $[n]$ be the integers from 1 to $n$. We denote by $Z \in \mathbb{R}^{n \times d}$ the feature matrix whose $i^{th}$ line corresponds to the vector $y_i x_i$. When not specified, $\|\cdot\|$ corresponds to any (definable) norm on $\mathbb{R}^d$. For a convex function $h$, $\partial h(\beta)$ denotes its subdifferential at $\beta$: $\partial h(\beta) = \{g \in \mathbb{R}^d : h(\beta') \geq h(\beta) + \langle g, \beta' - \beta \rangle, \forall \beta' \in \mathbb{R}^d\}$. For any scalar function $f : \mathbb{R} \to \mathbb{R}$ and vector $u \in \mathbb{R}^p$, the vector $f(u) \in \mathbb{R}^p$ corresponds to the component-wise application of $f$ over $u$. We denote by $\sigma : \mathbb{R}^n \to \mathbb{R}^n$ the softmax function equal to $\sigma(z) = \exp(z)/\sum_{i=1}^{n} \exp(z_i) \in \Delta_n$ where $\Delta_n$ is the unit simplex. For a convex potential $\phi$, we denote $D_\phi(\beta, \beta_0)$ the Bregman divergence equal to $\phi(\beta) - (\phi(\beta_0) + \langle \nabla \phi(\beta_0), \beta - \beta_0 \rangle) \geq 0$.

## 2 Problem set-up

We consider a dataset $(x_i, y_i)_{1 \leq i \leq n}$ with points $x_i \in \mathbb{R}^d$ and binary labels $y_i \in \{-1, 1\}$. We choose a loss function $\ell : \mathbb{R} \to \mathbb{R}$ and seek to minimise the empirical risk

$$L(\beta) = \sum_{i=1}^{n} \ell(y_i \langle x_i, \beta \rangle).$$

We propose to study the dynamics of mirror flow, which is the continuous-time limit of the *mirror descent* algorithm [Beck and Teboulle, 2003]. Mirror descent is a generalisation of gradient descent to non-Euclidean geometries induced by a given convex potential function $\phi : \mathbb{R}^d \to \mathbb{R}$. The method generates a sequence $(\hat{\beta}_k)_{k \geq 0}$ with $\hat{\beta}_0 = \beta_0 \in \mathbb{R}^d$ and

$$\nabla \phi(\hat{\beta}_{k+1}) = \nabla \phi(\hat{\beta}_k) - \gamma \nabla L(\hat{\beta}_k).$$

When the step size $\gamma$ goes to 0, the mirror descent iterates approach the solution $(\beta_t)_{t \geq 0}$ to the following differential equation:

$$\mathrm{d}\nabla\phi(\beta_t) = -\nabla L(\beta_t)\mathrm{d}t, \tag{MF}$$

initialised at $\beta_0$. Studying the mirror flow (MF) leads to simpler computations than its discrete counterpart, and still allows to obtain rich insights about the algorithm's behaviour.

We now state our standing assumptions on the loss function $\ell$ and potential $\phi$.

**Assumption 1.** *The loss $\ell$ satisfies:*

1. *$\ell$ is convex, twice continuously differentiable, decreasing and $\lim_{z \to +\infty} \ell(z) = 0$.*

2. *$\ell$ has an exponential tail, in the sense that $\ell(z) \underset{z \to \infty}{\sim} -\ell'(z) \underset{z \to \infty}{\sim} \exp(-z)$.*

The first part of the assumptions is very general and ensures that the empirical loss $L$ can be minimised 'at infinity'. The exponential tail is crucial: it enables to identify a unique maximum margin solution towards which the iterates converge in direction, independently of the considered loss. Both the exponential $\ell(z) = \exp(-z)$ and the logistic loss $\ell(z) = \ln(1 + \exp(-z))$ satisfy the conditions. On the other hand, losses with polynomial tails do not satisfy the second criterion. Similar assumptions on the tail appear when investigating the implicit bias of gradient descent for separable data [Soudry et al., 2018, Nacson et al., 2019b, Ji et al., 2020, Ji and Telgarsky, 2021, Chizat and Bach, 2020].

**Assumption 2.** *The potential $\phi : \mathbb{R}^d \to \mathbb{R}$ satisfies:*

1. *$\phi$ is twice continuously differentiable, strictly convex and coercive.*

2. *for every $c \in \mathbb{R}_{\geq 0}$ and $\beta_2 \in \mathbb{R}^d$, the sub-level set $\{\beta_1 \in \mathbb{R}^d, D_\phi(\beta_2, \beta_1) \leq c\}$ is bounded.*

3. *$\nabla^2\phi(\beta)$ is positive-definite for all $\beta \in \mathbb{R}^d$.*

4. *$\nabla\phi$ diverges at infinity: $\lim_{\|\beta\| \to \infty} \|\nabla\phi(\beta)\| = +\infty$.*

The first two points of the assumption are commonly used to ensure well-posedness of mirror descent [Bauschke et al., 2017]. The third one is necessary in continuous time to ensure the existence and uniqueness over $\mathbb{R}_{\geq 0}$ of a solution to the (MF) differential equation (in particular, we want to avoid the solution *"blowing up in finite time"*; see Lemma 2 in Appendix A). The coercive gradient assumption is crucial for our main result and we discuss it in more depth in Section 6.

Finally, we assume that the dataset is linearly separable.

**Assumption 3.** *There exists $\beta^\star \in \mathbb{R}^d$ such that $y_i\langle\beta^\star, x_i\rangle > 0$ for every $i \in [n]$.*

Notice that such $\beta^\star$'s correspond to minimisation directions: $L(\lambda\beta^\star) \overset{\lambda \to \infty}{\longrightarrow} 0$. Under the three previous assumptions, we can show that the mirror flow iterates $(\beta_t)_{t \geq 0}$ minimise the loss while diverging to infinity.

**Proposition 1.** *Considering the mirror flow $(\beta_t)_{t \geq 0}$, the loss converges towards $0$ and the iterates diverge: $\lim_{t \to \infty} L(\beta_t) = 0$ and $\lim_{t \to \infty} \|\beta_t\| = +\infty$.*

The proof relies on classical techniques used to analyse gradient methods in continuous time and we defer the proof to Appendix A. We now turn to the main question addressed in this paper:

*Among all minimising directions $\beta^\star$, towards which does the mirror flow converge?*

We initially offer a heuristic and intuitive answer to this question, setting the stage for the formal construction of the implicit regularisation problem.

## 3 Intuitive construction of the implicit regularisation problem

In this section, we give an informal presentation and proof sketch of our main result. A fully rigorous exposition is then provided in Section 4.

**Preliminaries.** Assume here for simplicity that $\ell(z) = \exp(-z)$. The mirror flow then writes

$$\frac{\mathrm{d}}{\mathrm{d}t} \nabla \phi(\beta_t) = L(\beta_t) \cdot Z^T q(\beta_t),$$

with $q(\beta_t) = \sigma(-Z\beta_t)$, where $\sigma$ is the softmax function and $Z$ the matrix with rows $(y_i x_i)_{i \in [n]}$. Note that $q(\beta_t)$ belongs to the unit simplex $\Delta_n$.

We simplify the differential equation by performing a time rescaling, which does not change the asymptotical behaviour. As $\theta : t \mapsto \int_0^t L(\beta_s) \mathrm{d}s$ is a bijection in $\mathbb{R}_{\geq 0}$ (see Lemma 4), we can speed up time and consider the accelerated iterates $\tilde{\beta}_t = \beta_{\theta^{-1}(t)}$. [1] By the chain rule, we have

$$\frac{\mathrm{d}}{\mathrm{d}t} \nabla \phi(\tilde{\beta}_t) = Z^\top q(\tilde{\beta}_t),$$

and therefore

$$\frac{1}{t} \nabla \phi(\tilde{\beta}_t) = \frac{1}{t} \nabla \phi(\beta_0) + Z^\top \left( \frac{1}{t} \int_0^t q(\tilde{\beta}_s) \mathrm{d}s \right). \tag{1}$$

From now on, we drop the tilde notation and assume that a change of time scale has been done. We want to characterise the directional limit of the diverging iterates $\beta_t$. To do so, we study their normalisation $\bar{\beta}_t := \frac{\beta_t}{\|\beta_t\|}$. As they form a bounded sequence, and $q(\beta_t) \in \Delta_n$ is also bounded, we can extract a subsequence[2] $(\bar{\beta}_{t_s}, q(\beta_{t_s}))_{s \in \mathbb{N}}$, with $\lim_{s \to \infty} t_s = \infty$ converging to some limit $(\bar{\beta}_\infty, q_\infty)$. By the Césaro average property, $\frac{1}{t_s} \int_0^{t_s} q(\beta_s) \mathrm{d}s$ also converges towards $q_\infty$. Equation (1) then yields

$$\frac{1}{t_s} \nabla \phi(\beta_{t_s}) \xrightarrow[s \to \infty]{} Z^\top q_\infty. \tag{2}$$

Observe that $q(\beta_t) = \sigma(-Z\beta_t)$ and the softmax function $\sigma$ approaches the argmax operator at infinity. Hence, as $\beta_t$ diverges, we expect that $q(\beta_t)_k \to 0$ for coordinates $k$ for which $(-Z\beta_t)_k$ is not maximal, *i.e.* $(Z\beta_t)_k$ not minimal. This observation is made formal in the following lemma. Its proof is straightforward and is given in Appendix A.

**Lemma 1.** *Assume that* $(\bar{\beta}_{t_s}, q(\beta_{t_s})) \xrightarrow{s \to \infty} (\bar{\beta}_\infty, q_\infty)$. *It holds that:*

$$(q_\infty)_k = 0 \quad \text{if} \quad y_k \langle x_k, \bar{\beta}_\infty \rangle > \min_{1 \leq i \leq n} y_i \langle x_i, \bar{\beta}_\infty \rangle.$$

In words, coordinates of $q_\infty$ which do not correspond to support vectors of $\bar{\beta}_\infty$ must be zero. Our goal is now to uniquely characterise $\bar{\beta}_\infty$ as the solution of a maximum margin problem.

### 3.1 Warm-up: gradient flow

As a warm-up, let us consider standard gradient flow, which corresponds to mirror flow with potential $\phi = \|\cdot\|_2^2/2$. In this case, Equation (2) becomes $\beta_{t_s}/t_s \to Z^\top q_\infty$. Since the normalised iterates satisfy $\bar{\beta}_{t_s} \to \bar{\beta}_\infty$, we get

$$\bar{\beta}_\infty = \frac{Z^\top q_\infty}{\|Z^\top q_\infty\|_2}.$$

Now notice that this equation along with the slackness conditions from Lemma 1 exactly correspond to the optimality conditions of the following convex minimisation problem:

$$\min_{\bar{\beta}} \|\bar{\beta}\|_2 \quad \text{under the constraint} \quad \min_{i \in [n]} y_i \langle x_i, \bar{\beta} \rangle \geq 1. \tag{3}$$

---

[1] $\tilde{\beta}_t$ can also be seen as the mirror flow trajectory but on the log-sum-exp function instead of the sum-exp function

[2] More precisely, we let $(t_n)_{n \in \mathbb{N}}$ be any sequence with $t_n \to \infty$ as $n \to \infty$ and then consider the discrete sequence $(\bar{\beta}_{t_n}, q(\beta_{t_n}))_{n \in \mathbb{N}}$. We can then apply the Bolzano–Weierstrass theorem to extract a convergent subsequence. We will show that the limit $(\bar{\beta}_\infty, q_\infty)$ is unique and does not depend on the sequence $(t_n)$, and therefore the continuous-time process $(\bar{\beta}_t, q(\beta_t))_{t \in \mathbb{R}}$ must converge towards it.

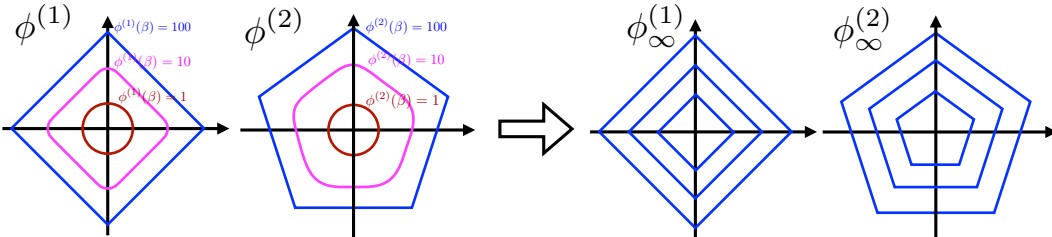

Figure 2: *Left two:* Sketch of the level lines of two different potentials $\phi^{(1)}, \phi^{(2)} : \mathbb{R}^2 \to \mathbb{R}$. *Right two:* Their corresponding horizon functions $\phi_\infty^{(1)}, \phi_\infty^{(2)}$ as defined in Section 4.1.

Furthermore, the $\ell_2$-unit ball being strictly convex, Problem (3) has a unique solution to which $\bar{\beta}_\infty$ must therefore be equal. Importantly, notice that Problem (3) uniquely defines the limit of **any extraction** on the normalised iterates $\bar{\beta}_t$: the normalised iterates $\bar{\beta}_t$ must therefore converge towards the $\ell_2$-maximum margin. We recover the implicit regularisation result from Soudry et al. [2018]:

$$\bar{\beta}_\infty = \underset{\min_i y_i \langle x_i, \bar{\beta} \rangle \geq 1}{\arg \min} \|\bar{\beta}\|_2.$$

### 3.2 General potential: introducing the horizon function $\phi_\infty$

We now tackle general potentials $\phi$. In the general case, the challenge of identifying the max-margin problem to which the iterates converge in direction stems from the fact that if the potential $\phi$ is not $L$-homogeneous[3], its geometry changes as the iterates diverge. More precisely, its sub-level sets $S_c := \{\beta \in \mathbb{R}^d, \phi(\beta) \leq c\}$ change of shape as $c$ increase, as illustrated by Figure 2 (Left).

However, we can hope that these sets have a **limiting shape** at infinity, meaning that the normalised sub-level sets $\bar{S}_c := S_c / R_c$ where $R_c := \max_{\beta \in S_c} \|\beta\|$ converge to some limiting convex set $S_\infty$ as $c \to \infty$. We can then construct an asymmetric norm[4] $\phi_\infty$ which has $S_\infty$ as its unit ball. **In words, $\phi_\infty$ captures the shape of $\phi$ at infinity.** This informal construction is made rigorous in Section 4.1. We state here the crucial consequence of this construction.

**Corollary 1.** *The horizon function $\phi_\infty$ is such that for any sequence $\beta_t$ diverging to infinity for which $\frac{\beta_t}{\|\beta_t\|}$ and $\frac{\nabla \phi(\beta_t)}{\|\nabla \phi(\beta_t)\|}$ both converge, then:*

$$\lim_{t \to \infty} \frac{\nabla \phi(\beta_t)}{\|\nabla \phi(\beta_t)\|} \in \lambda \cdot \partial \phi_\infty(\bar{\beta}_\infty), \quad \text{where} \quad \bar{\beta}_\infty = \lim_{t \to \infty} \frac{\beta_t}{\|\beta_t\|},$$

*for some strictly positive factor $\lambda$.*

Using this construction, we can derive the optimality conditions satisfied by $\bar{\beta}_\infty$. From the convergence in Equation (2) and that of $\bar{\beta}_t \to \bar{\beta}_\infty$, applying Corollary 1, we obtain that:

$$Z^\top q_\infty \in \lambda \cdot \partial \phi_\infty(\bar{\beta}_\infty).$$

Up to a positive multiplicative factor (which is irrelevant due to the positive homogeneity of the quantities involved), this condition along with Lemma 1 are exactly the optimality conditions of the convex problem

$$\underset{\bar{\beta} \in \mathbb{R}^d}{\min} \quad \phi_\infty(\bar{\beta}) \quad \text{under the constraint} \quad \underset{i \in [n]}{\min} \; y_i \langle x_i, \bar{\beta} \rangle \geq 1.$$

The limiting direction $\bar{\beta}_\infty$ must therefore belong to the set of its solutions. Assuming that this set contains a single element of norm 1 (we refer to the next section for comments concerning the uniqueness), we deduce that the iterates $\bar{\beta}_t$ must converge towards it:

$$\lim_{t \to \infty} \frac{\beta_t}{\|\beta_t\|} \propto \underset{\min_i y_i \langle x_i, \bar{\beta} \rangle \geq 1}{\arg \min} \; \phi_\infty(\bar{\beta}).$$

---

[3]A function is $L$-homogeneous if there exists $L > 0$ such that $\phi(c\beta) = c^L \phi(\beta)$ for all $\beta$ and $c > 0$

[4]An asymmetric norm $p$ satisfies all the properties of a norm except the symmetry equality $p(-\beta) = p(\beta)$

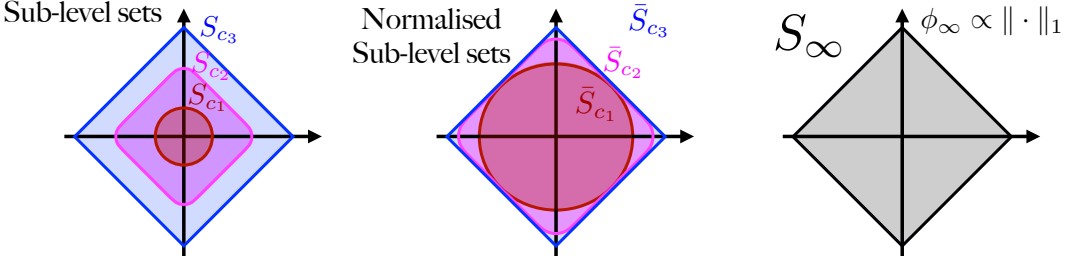

Figure 3: Illustration of the construction of the horizon shape $S_\infty$. *Left:* the sub-level sets $S_c$ change of shape and are increasing. *Middle:* in order to avoid the shapes blowing up, we normalise them to keep them in the unit ball (here we choose the arbitrary constraining norm to be the $\ell_1$-norm). *Right:* the normalised sub-level sets $\bar{S}_c$ converge to a limiting set $S_\infty$ for the Hausdorff distance.

## 4 Main result: directional convergence towards the $\phi_\infty$-max margin

We now state our formal results, starting with the precise construction of the horizon function $\phi_\infty$, followed by the theorem showing convergence of the iterates towards the $\phi_\infty$-max-margin.

### 4.1 Construction of the horizon function $\phi_\infty$

We first define the **horizon shape** of a potential $\phi$, and provide sufficient conditions for its existence. Then, we use this shape to construct a **horizon function** $\phi_\infty$, which allows the interpretation of the directional limits of gradients of $\phi$ at infinity. The proofs require technical elements from variational analysis to ensure that the limits are well-defined; these are deferred to Appendix B.

**Horizon shape.** Assume w.l.o.g. that $\phi(0) = 0$. For $c \geq 0$, consider the sublevel set:
$$S_c(\phi) = \{\beta \in \mathbb{R}^d \,:\, \phi(\beta) \leq c\},$$
which is nonempty and compact by coercivity of $\phi$. We can then define the normalised sublevel set:
$$\bar{S}_c = \frac{1}{R_c} S_c, \quad R_c = \max\{\|\beta\| \,:\, \beta \in S_c\}. \tag{4}$$

By construction, the set $\bar{S}_c$ belongs to the unit ball. We are interested in the limit of $\bar{S}_c$ as $c \to \infty$.

**Definition 1.** *We say that $\phi$ admits a horizon shape if the family of normalized sublevel sets $(\bar{S}_c)_{c>0}$ defined in Equation (4) converges to some compact set $S_\infty$ as $c \to \infty$ for the Hausdorff distance. In addition, we say that this shape is non-degenerate if the origin belongs to the interior of $S_\infty$.*

The Hausdorff distance is a natural distance on compact sets [see Rockafellar and Wets, 1998, Section 4.C., for a definition]. In Proposition 2, we prove the existence of the horizon shape for a large class of functions which contains all the potentials with domain $\mathbb{R}^d$ encountered in practice. Although the horizon shape is guaranteed to exist for most functions, we cannot a priori prove that it is non-degenerate, as the normalized sub-levels $\bar{S}_c$ can become 'flat' as $c \to \infty$.[5] Given the technical complexity associated with this case, we now focus exclusively on non-degenerate horizon shapes.

**Horizon function.** If $\phi$ admits a non-degenerate horizon shape $S_\infty$, we define its horizon function as the *Minkowski gauge* [Rockafellar and Wets, 1998, Section 11.E] of $S_\infty$:
$$\phi_\infty(\bar{\beta}) := \inf\left\{r > 0 \,:\, \frac{\bar{\beta}}{r} \in S_\infty\right\} \quad \text{for } \bar{\beta} \in \mathbb{R}^d.$$

By construction, the horizon function $\phi_\infty$ is an asymmetric norm and its sub-level sets correspond to scaled versions of $S_\infty$ [see Rockafellar and Wets, 1998, Section 11.C, for more properties]. For example, in the case of the horizon shape $S_\infty$ illustrated in Figure 3, the corresponding horizon function $\phi_\infty$ is proportional to the $\ell_1$-norm. Although the construction of $\phi_\infty$ presented here is rather abstract, we show in Theorem 3 that for separable potentials defined over $\mathbb{R}^d$, it can be computed with an explicit formula. Though different, our definition of the horizon function shares many similarities with the classical concept of horizon function from convex analysis [Rockafellar and Wets, 1998]. We discuss the links between the two notions at the end of Section 4.3.

---

[5]Consider for instance $\phi(x, y) = x^2 + y^4$ on $\mathbb{R}^2$, for which the horizon shape is $[-1, 1] \times \{0\}$.

## 4.2 Main result: directional convergence of the iterates towards the $\phi_\infty$-max-margin

We can now state our main result which fully characterises the directional convergence of mirror flow.

**Theorem 2.** *Assume that $\phi$ admits a non-degenerate horizon shape and let $\phi_\infty$ be its horizon function. Assuming that the following $\phi_\infty$-max-margin problem has a unique minimiser, then the mirror flow normalised iterates $\bar{\beta}_t = \frac{\beta_t}{\|\beta_t\|}$ converge towards a vector $\bar{\beta}_\infty$ and*

$$\bar{\beta}_\infty \propto \underset{\bar{\beta} \in \mathbb{R}^d}{\arg\min}\, \phi_\infty(\bar{\beta}) \quad \text{under the constraint} \quad \min_{i \in [n]}\, y_i \langle x_i, \bar{\beta} \rangle \geq 1,$$

*where the symbol $\propto$ denotes positive proportionality.*

**Remark on the uniqueness of the margin problem.** If the unit ball of $\phi_\infty$ is strictly convex, then the $\phi_\infty$-max-margin problem has a unique solution. However, in the general case, there may exist an infinity of solutions and weak but ad hoc assumptions on the dataset are required to guarantee uniqueness. For instance, if $\phi_\infty$ is proportional to the $\ell_1$-norm, a common assumption which ensures uniqueness is assuming that the data features are in *general position* [Dossal, 2012].

## 4.3 Assumptions guaranteeing the existence of $\phi_\infty$

Our main result, presented in Theorem 2 relies on the existence of a horizon shape, $S_\infty$, as described in Definition 1. From this shape, the asymmetric norm $\phi_\infty$ is constructed.

We show here that the existence of $S_\infty$ is ensured for a large class of 'nice' functions, specifically those *definable in o-minimal structures* [Dries, 1998]. For the reader unfamiliar with this notion, this class contains all 'reasonable' functions used in practice, such as polynomials, logarithms, exponentials, and 'reasonable' combinations of those. This is a typical assumption used for instance to prove the convergence of optimisation methods through the Kurdyka–Łojasiewicz property [Attouch et al., 2011].

**Proposition 2.** *If any of the three following conditions hold: (i) $\phi$ is a finite composition of polynomials, exponentials and logarithms, (ii) $\phi$ is globally subanalytic, (iii) $\phi$ is definable in a o-minimal structure on $\mathbb{R}$; then $\phi$ admits a horizon shape $S_\infty$.*

The proof is technical and we defer it to Appendix B. Although the previous proposition ensures the existence $\phi_\infty$ for a wide range of potentials, it does not offer a direct method for computing it. In the following, we show that for potentials that are both separable and even, a simple formula exists, allowing for the direct calculation of $\phi_\infty$.

**Assumption 4.** *The potential $\phi$ is separable, in the sense that there exists $\varphi : \mathbb{R} \to \mathbb{R}_{\geq 0}$ such that $\phi(\beta) = \sum_{i=1}^d \varphi(\beta_i)$. We assume that $\varphi$ satisfies Assumption 2, that it is definable in a o-minimal structure on $\mathbb{R}$ and that it is an even function. W.l.o.g. we assume that $\varphi(0) = 0$.*

We note that $\varphi$ is a bijection over $\mathbb{R}_{\geq 0}$, and denote by $\varphi^{-1}$ its inverse. We consider the function $\varphi^{-1} \circ \phi$, which can be seen as a renormalisation of $\phi$. It has the same level sets as $\phi$ and ensures that $\lim_{\eta \to 0} \eta \varphi^{-1}(\phi(\bar{\beta}/\eta))$ exists in $\mathbb{R}_{>0}$ for all $\bar{\beta}$. These two observations lead to the following theorem.

**Theorem 3.** *Under Assumption 4, the potential $\phi$ admits a non-degenerate horizon shape and its horizon function is such that there exists $\lambda > 0$ such that for every $\bar{\beta} \in \mathbb{R}^d$:*

$$\phi_\infty(\bar{\beta}) = \lambda \lim_{\eta \to 0} \eta \cdot \varphi^{-1}\left(\phi\left(\frac{\bar{\beta}}{\eta}\right)\right).$$

We use this simple formula when computing $\phi_\infty$ for various potentials in the next section.

**Remark on previous notions of horizon function.** In the convex analysis literature, the horizon function is typically defined as $\phi_\infty(\bar{\beta}) = \lim_{\eta \to 0} \eta \phi(\bar{\beta}/\eta)$ [Rockafellar and Wets, 1998, Laghdir and Volle, 1999]. In our context, this definition would yield a function which equals $+\infty$ everywhere except at the origin. In contrast, our definition ensures that $\phi_\infty$ attains finite values over $\mathbb{R}^d$. The distinction stems from our way of normalising the level sets by $R_c$ in Section 4.1, or alternatively, from the composition by $\varphi^{-1}$ in the separable case. The two constructions would coincide only if $\phi$ was Lipschitz continuous, which is at odds with Assumption 2.

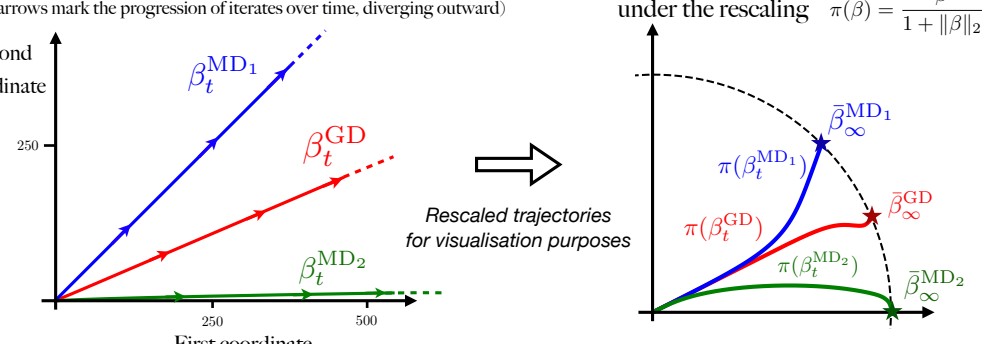

Figure 4: Mirror flow trajectories on a 2-dimensional dataset for three different potentials (exact same setting as in Figure 1). *Left:* the iterates diverge to infinity and the directional convergence depends on the choice of potential. *Right:* the normalised iterates converge towards their respective $\phi_\infty$-maximum-margin predictors (illustrated by stars), as predicted by Theorem 2.

## 5    Applications and experiments

In this section, we illustrate our main result using various potentials.

**Homogeneous potentials.**    We first consider potentials $\phi$ which are $L$-homogeneous, i.e., there exists $L > 0$ such that for all $c > 0$ and $\beta \in \mathbb{R}^d$, $\phi(c\beta) = c^L \phi(\beta)$. In this case, the sublevel sets $\bar{S}_c$ are all equal. It follows that $\phi_\infty \propto \phi^{1/L}$. An important example is the case of $\phi = \| \cdot \|_p^p$ where $\| \cdot \|_p$ corresponds to the $\ell_p$-norm with $p > 1$, for which we get that $\phi_\infty \propto \| \cdot \|_p$ and we recover the result from Sun et al. [2022, 2023].

**Hyperbolic-cosine entropy potential.**    Finally, we consider $\phi^{\mathrm{MD}_1}(\beta) = \sum_{i=1}^d (\cosh(\beta_i) - 1)$. Applying Theorem 3, we get that $\phi_\infty \propto \| \cdot \|_\infty$.

**Hyperbolic entropy potential.**  The  hyperbolic  entropy  potential:    $\phi^{\mathrm{MD}_2}(\beta) = \sum_{i=1}^d (\beta_i \mathrm{arcsinh}(\beta_i) - \sqrt{\beta_i^2 + 1} - 1)$  plays  a  central  role  in  works  considering  diagonal linear networks [Woodworth et al., 2020, Pesme and Flammarion, 2023]. Applying Theorem 3, we obtain that $\phi_\infty \propto \| \cdot \|_1$ and we recover the result from Lyu and Li [2020] and Moroshko et al. [2020].

**Experimental details concerning Figure 1.**    As shown in Figure 1 (Middle), we generate 40 points with positive labels and 40 points with negative labels. Starting from $\beta_0 = 0$, we run mirror descent with the exponential loss $\ell(z) = \exp(-z)$ and with the three following potentials:

$$(i) \ \phi^{\mathrm{GD}} = \| \cdot \|_2^2, \qquad (ii) \ \phi^{\mathrm{MD}_1} = \mathrm{cosh\text{-}entropy}, \qquad (iii) \ \phi^{\mathrm{MD}_2} = \mathrm{Hyperbolic\ entropy}.$$

We first observe in Figure 1 (Left) that the training loss converges to zero, as predicted by Proposition 1, with a convergence rate that varies across different potentials. Moreover, as illustrated in Figure 1 (Middle and Right), the iterates converge in direction towards their respective unique $\phi_\infty$-max margin solutions associated with the following geometries:

$$(i) \ \phi_\infty^{\mathrm{GD}} \propto \| \cdot \|_2, \qquad (ii) \ \phi_\infty^{\mathrm{MD}_1} = \| \cdot \|_\infty, \qquad (iii) \ \phi_\infty^{\mathrm{MD}_2} \propto \| \cdot \|_1.$$

Therefore, by employing various potentials, we can induce different implicit biases, leading to distinct generalisation properties depending on the data distribution. The trajectories of the mirror descent iterates are shown and commented in Figure 4.

# 6 Conclusion and limitations

In this paper, we offer a comprehensive characterisation of the implicit bias of mirror flow for separable classification problems. This characterisation is framed in terms of the horizon function associated with the mirror descent potential, leveraging the asymptotic geometry induced by the potential. Note that we did not cover the **discrete** mirror descent algorithm; we believe the analysis would extend without additional difficulties compared to the continuous counterpart.

**Extensions and open problems.** Our results being purely asymptotic, characterising the rate at which the normalised iterates converge towards the maximum-margin solution is an open direction for future research. Furthermore, we note that our analysis does not cover potentials that are defined only on a strict subset of $\mathbb{R}^d$ (such as the log-barrier and the negative entropy), and with possibly non-coercive gradients. This class of potentials is of interest as it arises when investigating deep architectures, such as diagonal linear networks of depth $D > 2$. In this setting, it is known that gradient flow on the weights lead to a mirror flow on the predictors with a certain potential $\phi_D$ [Woodworth et al., 2020]. Interestingly, the potentials $\phi_D$ have non-coercive gradients and their horizon functions do not depend on the depth $D$ as they are all proportional to the $\ell_1$-norm. The predictors are, however, known to converge in direction towards a KKT point of the non-convex $\ell_{2/D}$-max-margin problem [Lyu and Li, 2020] which can be different from the $\ell_1$-max-margin problem [Moroshko et al., 2020]. This observation highlights that our coercive gradient assumption is necessary for our result to hold. However, extending our analysis beyond this assumption is a promising direction for understanding gradient dynamics in deep architectures.

## Acknowledgments and Disclosure of Funding

S.P. would like to thank Loucas Pillaud-Vivien for the helpful discussions at the beginning of the project as well as Pierre Quinton for the careful proofreading of the paper. The authors also thank Jérôme Bolte and Edouard Pauwels for the precious help on the existence of the Hausdorff limit of definable families, as well as Nati Srebro and Lénaïc Chizat for the remarks on the limitations of our result for potentials with non-coercive gradients. This work was supported by the Swiss National Science Foundation (grant number 212111).

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

# A   Proofs of properties of the mirror flow in the classification setting

We start by proving the existence and uniqueness of (MF) over $\mathbb{R}_{\geq 0}$. The proof is standard and relies on ensuring that the iterates do not diverge in finite time.

**Lemma 2.** *For any initialisation $\beta_0 \in \mathbb{R}^d$, there exists a unique solution defined over $\mathbb{R}_{\geq 0}$ which satisfies* (MF) *for all $t \geq 0$ and with initial condition $\beta_{t=0} = \beta_0$.*

*Proof.* From Assumption 2, we have that $\phi$ is differentiable, strictly convex and its gradient is coercive. Consequently, $\nabla \phi$ is bijective over $\mathbb{R}^d$ (see Rockafellar [1970], Theorem 26.6). Furthermore, the Fenchel conjugate $\phi^*$ is differentiable over $\mathbb{R}^d$ and $(\nabla \phi)^{-1} = \nabla \phi^*$.

To prove the existence and uniqueness of a global solution of (MF), we first consider the following differential equation:

$$\mathrm{d}u_t = -\nabla L(\nabla \phi^*(u_t))\mathrm{d}t, \tag{5}$$

with initial condition $u_{t=0} = \nabla \phi^*(\beta_0)$.

Since $L$ is $\mathcal{C}^2$, $\nabla L$ is Lipschitz on all compact sets. Furthermore, since $\nabla^2 \phi$ is p.s.d., $\nabla \phi^* = (\nabla \phi)^{-1}$ is $\mathcal{C}^1$ and therefore Lipschitz on all compact sets. Hence $\nabla L \circ \nabla \phi^*$ is Lipschitz on all compact sets and from the Picard-Lindelöf theorem, there exists a unique maximal (*i.e.* which cannot be extended) solution $(u_t)$ satisfying eq. (5) such that $u_{t=0} = \nabla \phi^*(\beta_0)$. We denote $[0, T_{\max})$ the intersection of this maximal interval of definition (which must be open) and $\mathbb{R}_{\geq 0}$. Our goal is now to prove that $T_{\max} = +\infty$. To do so, we assume that $T_{\max}$ is finite and we will show that this leads to a contradiction due to the fact that the iterates $\beta_t$ cannot diverge in finite time. Let $\beta_t := \nabla \phi^*(u_t)$ and notice that $\beta_t$ is therefore the unique solution satisfying (MF) over $[0, T_{\max})$ with $\beta_{t=0} = \beta_0$.

**Bounding the trajectory of $\beta_t$ over $[0, T_{\max})$.** Pick any $\beta \in \mathbb{R}^d$ and notice that by convexity of $L$:

$$\frac{\mathrm{d}}{\mathrm{d}t} D_\phi(\beta, \beta_t) = -\langle \nabla L(\beta_t), \beta_t - \beta \rangle \leq -(L(\beta_t) - L(\beta)) \leq L(\beta) - L_{\min}.$$

Where $L_{\min}$ is a lower bound on the loss. Integrating from 0 to $t < T_{\max}$ we get:

$$D_\phi(\beta, \beta_t) \leq t \cdot (L(\beta) - L_{\min}) + D_\phi(\beta, \beta_0)$$
$$\leq T_{\max} \cdot (L(\beta) - L_{\min}) + D_\phi(\beta, \beta_0)$$

Therefore, due to Assumption 2, the iterates $\beta_t$ are bounded over $[0, T_{\max})$. The proof from here is standard (see e.g. Attouch et al. [2000], Theorem 3.1): from eq. (5) we get that $\dot{u}_t$ is bounded over $[0, T_{\max})$ and $\sup_{t \in [0, T_{\max})} \|\dot{u}_t\| =: C < +\infty$ which means that $\|u_t - u_{t'}\| \leq C|t - t'|$. Hence $\lim_{t \to T_{\max}} u_t =: u_\infty$ must exist. Applying the Picard-Lindelöf again at time $T_{\max}$ with initial condition $u_\infty$ violates the initial maximal interval assumption. Therefore $T_{\max} = +\infty$ which concludes the proof. $\square$

We now recall and prove classical results on the mirror flow in the classification setting.

**Proposition 1.** *Considering the mirror flow $(\beta_t)_{t \geq 0}$, the loss converges towards $0$ and the iterates diverge:* $\lim_{t \to \infty} L(\beta_t) = 0$ *and* $\lim_{t \to \infty} \|\beta_t\| = +\infty$.

*Proof.* **The loss is decreasing.** $\frac{\mathrm{d}}{\mathrm{d}t} L(\beta_t) = \langle \nabla L(\beta_t), \dot{\beta}_t \rangle = -\langle \nabla^2 \phi(\beta_t)^{-1} \nabla L(\beta_t), \nabla L(\beta_t) \rangle \leq 0$, where the inequality is due to the convexity of the potential $\phi$.

**Convergence of the loss towards $0$.** Now consider the Bregman divergence between an arbitrary point $\beta$ and $\beta_t$:

$$D_\phi(\beta, \beta_t) = \phi(\beta) - \phi(\beta_t) - \langle \nabla \phi(\beta_t), \beta - \beta_t \rangle \geq 0.$$

which is such that:

$$\frac{\mathrm{d}}{\mathrm{d}t} D_\phi(\beta, \beta_t) = \langle \frac{\mathrm{d}}{\mathrm{d}t} \nabla \phi(\beta_t), \beta_t - \beta \rangle$$
$$= -\langle \nabla L(\beta_t), \beta_t - \beta \rangle$$
$$\leq -(L(\beta_t) - L(\beta)) \tag{6}$$

where the inequality is by convexity of the loss. Integrating and due to the decrease of the loss, we get that:

$$L(\beta_t) \leq \frac{1}{t} \int_0^t L(\beta_s) \, \mathrm{d}s$$

$$\leq L(\beta) + \frac{D_\phi(\beta, \beta_0) - D_\phi(\beta, \beta_t)}{t}$$

$$\leq L(\beta) + \frac{D_\phi(\beta, \beta_0)}{t}$$

Since this is true for all point $\beta$, we get that $L(\beta_t) \leq \inf_{\beta \in \mathbb{R}^d} L(\beta) + \frac{D_\phi(\beta, \beta_0)}{t}$. It remains to show that the right hand term goes to 0 as $t$ goes to infinity. To show this, let $\varepsilon > 0$, by the separability assumption we get that there exists $\beta^\star$ such that $\min y_i \langle x_i, \beta^\star \rangle > 0$. Since $L(\lambda \beta_\star) \underset{\lambda \to \infty}{\longrightarrow} 0$, we can choose $\lambda$ big enough such that $L(\lambda \beta^\star) < \varepsilon$ and then $t_\lambda$ large enough such that $\frac{1}{t_\lambda} D_\phi(\lambda \beta^\star, \beta_0) < \varepsilon$. The loss therefore converges to 0.

**Divergence of the iterates.**

For all $i \in [n]$, $\ell(y_i \langle x_i, \beta_t \rangle) \leq L(\beta_t) \underset{t \to \infty}{\longrightarrow} 0$. Due to the assumptions on the loss, this translates into $y_i \langle x_i, \beta_t \rangle \underset{t \to \infty}{\longrightarrow} \infty$, hence $\|\beta_t\| \underset{t \to \infty}{\longrightarrow} +\infty$.

$\square$

**Introducing a few notations.** Before giving two important lemmas, we provide a few notations. Recall that $Z$ is the data matrix of size $n \times d$ whose $i^{th}$ row is $y_i x_i$, we then have that $\nabla L(\beta) = Z^\top \ell'(Z\beta)$, where $\ell'$ is applied component wise. We now denote by $q(\beta)$ the vector in $\mathbb{R}^n$ equal to:

$$q(\beta) = \frac{\ell'(Z\beta)}{\ell'(\ell^{-1}(\sum_i \ell(y_i \langle x_i, \beta \rangle)))}$$

Notice that due to $\ell > 0$, $\ell' < 0$, $\ell^{-1}$ is increasing and $\ell'$ is decreasing, we have that $q(\beta) > 0$ and that for all $i_0 \in [n]$,

$$q(\beta)_{i_0} = \frac{\ell'(y_{i_0} \langle x_{i_0}, \beta \rangle)}{\ell'(\ell^{-1}(\sum_i \ell(y_i \langle x_i, \beta \rangle)))} \leq \frac{\ell'(y_{i_0} \langle x_{i_0}, \beta \rangle)}{\ell'(\ell^{-1}(\ell(y_{i_0} \langle x_{i_0}, \beta \rangle)))} \leq 1.$$

Therefore $q(\beta) \in (0, 1]^n$.

We further denote

$$a_t := -\ell'(\ell^{-1}(\sum_i \ell(y_i \langle x_i, \beta_t \rangle))) > 0.$$

This way we can simply write $\nabla L(\beta_t) = -a_t Z^\top q_t$ with $q_t := q(\beta_t)$.

Integrating the flow (MF), we write:

$$\nabla \phi(\beta_t) = \nabla \phi(\beta_0) - \int_0^t \nabla L(\beta_s) \mathrm{d}s$$

$$= \nabla \phi(\beta_0) + Z^\top \int_0^t a_s q_s \mathrm{d}s. \tag{7}$$

**Two lemmas.** In the following lemma we recall and prove that a coordinate $q_\infty[k]$ must be equal to 0 if datapoint $x_k$ is not a support vector of $\bar{\beta}_\infty$.

**Lemma 3.** *For some function $C_t \to \infty$, if the iterates $\bar{\beta}_t = \frac{\beta_t}{C_t}$ converge towards a vector which we denote $\bar{\beta}_\infty$ and $q_t$ converges towards a vector $q_\infty \in [0, 1]^n$. Then it holds that:*

$$q_\infty[k] = 0 \qquad if \qquad y_k \langle x_k, \bar{\beta}_\infty \rangle > \min_{1 \leq i \leq n} y_i \langle x_i, \bar{\beta}_\infty \rangle.$$

*In words, $q_\infty[k] = 0$ if $x_k$ is not a support vector.*

*Proof.* Recall that

$$q(\beta_t) = \frac{\ell'(Z\beta_t)}{\ell'(\ell^{-1}(\sum_i \ell(y_i\langle x_i, \beta_t\rangle)))}.$$

From Proposition 1, we have that $\min_{i\in[n]} y_i\langle x_i, \bar\beta_\infty\rangle > 0$ and we denote this margin as $\gamma$. Now consider $k \in [n]$ which is not a support vector, i.e, $y_k\langle x_k, \bar\beta_\infty\rangle > \min_{i\in[n]} y_i\langle x_i, \bar\beta_\infty\rangle$ and without loss of generality assume that $y_1\langle x_1, \bar\beta_\infty\rangle = \min_{1\le i\le n} y_i\langle x_i, \bar\beta_\infty\rangle$. We denote by $\delta = \langle y_k x_k - y_1 x_1, \bar\beta_\infty\rangle > 0$ the gap. Then

$$\begin{aligned}
q(\beta_t)_k &= \frac{\ell'(C_t\langle x_k, \bar\beta_t\rangle)}{\ell'(\ell^{-1}(\sum_i \ell(C_t y_i\langle x_i, \bar\beta_t\rangle)))} \\
&\le \frac{\ell'(C_t y_k\langle x_k, \bar\beta_t\rangle)}{\ell'(C_t y_1\langle x_1, \bar\beta_t\rangle)}
\end{aligned}$$

We write $\bar\beta_t = \bar\beta_\infty + r_t$ where $(r_t)_{t\ge0} \in \mathbb{R}^d$ converges to 0. For $t$ big enough, we have that $y_k\langle x_k, \bar\beta_t\rangle \ge y_k\langle x_k, \bar\beta_\infty\rangle - \frac{\delta}{4}$ and $y_1\langle x_1, \bar\beta_t\rangle \le y_1\langle x_1, \bar\beta_\infty\rangle + \frac{\delta}{4}$. Therefore for $t$ large enough, since $\ell'$ is negative and increasing:

$$\begin{aligned}
q(\beta_t)[k] &\le \frac{\ell'(C_t(y_k\langle x_k, \bar\beta_\infty\rangle - \delta/4))}{\ell'(C_t(y_1\langle x_1, \bar\beta_\infty\rangle + \delta/4))} \\
&\le \frac{\ell'(C_t(\gamma + \delta/2 + \delta/4))}{\ell'(C_t(\gamma + \delta/2))} \xrightarrow[t\to\infty]{} 0,
\end{aligned}$$

where the last term converge to 0 due to the exponential tail of $-\ell'$ and that $C_t \to \infty$. $\qquad\square$

We here reformulate and prove Lemma 1.

**Lemma 4** (Reformulation of Lemma 1). *Denoting* $a(\beta_t) := -\ell'(\ell^{-1}(\sum_i \ell(y_i\langle x_i, \beta_t\rangle))) > 0$, *we have that* $\int_0^t a(\beta_s)\mathrm{d}s \xrightarrow[t\to\infty]{} +\infty$. *For* $\ell(z) = \exp(-z)$, *this translates to* $\int_0^t L(\beta_s)\mathrm{d}s \to \infty$.

*Proof.* Recall that $\nabla\phi(\beta_t) = \nabla\phi(\beta_0) + Z^\top \int_0^t a(\beta_s)q(\beta_s)\mathrm{d}s$, therefore

$$\begin{aligned}
\|\nabla\phi(\beta_t)\| &\le \|\nabla\phi(\beta_0)\| + \sum_{i=1}^n \|x_i\| \int_0^t a(\beta_s)q(\beta_s)[i]\mathrm{d}s \\
&\le \|\nabla\phi(\beta_0)\| + \Big(\sum_{i=1}^n \|x_i\|\Big) \int_0^t a(\beta_s)\mathrm{d}s.
\end{aligned}$$

Where the first inequality is due to the triangle inequality and the second to the fact $q(\beta) \in (0,1]^n$. Since the iterates diverge, we have from Assumption 2 that $\|\nabla\phi(\beta_t)\| \xrightarrow[t\to\infty]{} \infty$ and therefore that $\int_0^t a(\beta_s)\mathrm{d}s \xrightarrow[t\to\infty]{} +\infty$. $\qquad\square$

# B  Differed proofs on the construction of $\phi_\infty$

As mentioned in the main text, the following property highlights the fact that all 'reasonable' potentials have a horizon shape.

**Proposition 2.** *If any of the three following conditions hold: (i) $\phi$ is a finite composition of polynomials, exponentials and logarithms, (ii) $\phi$ is globally subanalytic, (iii) $\phi$ is definable in a o-minimal structure on $\mathbb{R}$; then $\phi$ admits a horizon shape $S_\infty$.*

*Proof.* Note that points (i) and (ii) are particular cases of (iii) [Dries, 1998, Bolte et al., 2007]. If $h$ is definable in a o-minimal structure, then so is the sublevel set $S_c$ for $c > 0$, and so is the normalization factor $R_c$ since it can be defined in first-order logic as

$$R_c = \{r \in \mathbb{R} \,:\, \exists \beta^* \in S_c, \|\beta^*\| = r \text{ and } \forall \beta \in S_c, \|\beta\| \le r\}.$$

Therefore, $(\bar{S}_c)_{c>0}$ if a definable family of definable and compact sets. Then so is the family $(\bar{S}_{t^{-1}})_{t\in(0,1]}$. Since all the sets belong to the unit ball of $\mathbb{R}^d$, they lie in the sets of compact subsets of $B(0,1)$. This set is compact for the Hausdorff metric [Aliprantis and Border, 2006, Thm 3.85]; therefore, there exists a sequence $(t_k)_{k\in\mathbb{N}}$ such that $t_k \to 0$ and $(\bar{S}_{t_k^{-1}})_{k\in\mathbb{N}}$ converges to some set $\bar{S}$.

We can then apply Corollary 2 of Kocel-Cynk et al. [2014], which states that there exists a definable arc $\gamma : (0,1] \to (0,1]$ such that $\lim_{\tau\to 0} \gamma(\tau) = 0$ and $\bar{S} = \lim_{\tau\to 0} \bar{S}_{\gamma(\tau)^{-1}}$. This implies that the limit $\bar{S}$ is uniquely defined and therefore that $\lim_{t\to 0} \bar{S}_{t^{-1}} = \bar{S}$. $\qquad\square$

The next corollary is a more general restatement of Corollary 1. It shows that the construction of $\phi_\infty$ enables to take the limit $\lim_t \frac{\nabla\phi(\beta_t)}{t} \propto \partial\phi_\infty(\bar{\beta}_\infty)$.

**Corollary 2.** *Assume that $\phi$ admits a non-degenerate horizon shape $S_\infty$. Then its horizon function $\phi_\infty$ satisfies the following properties.*

1. *$\phi_\infty$ is convex and finite-valued on $\mathbb{R}^d$,*

2. *Let $(\beta_s)_{s>0}$ be a continous sequence such that when $s \to \infty$:*

   $(a)$ $\|\beta_s\| \to \infty$,   $(b)$ $\dfrac{\beta_s}{\|\beta_s\|} \to \bar{\beta}$ *for some* $\bar{\beta} \in \mathbb{R}^d$,   $(c)$ $\dfrac{\nabla\phi(\beta_s)}{\|\nabla\phi(\beta_s)\|} \to \bar{g}$ *for some* $\bar{g} \in \mathbb{R}^d$.

   *Then $\bar{g}$ is proportional to a subgradient of $\phi_\infty$ at $\bar{\beta}$:*

   $$\bar{g} \in \lambda\partial\phi_\infty(\bar{\beta}) \quad \text{for some } \lambda > 0.$$

*Proof.* The sequence of sets $(\bar{S}_c)$ is contained in the compact ball $B(0,1)$; therefore, Hausdorff convergence is equivalent to Painlevé-Kuratowski convergence [Rockafellar and Wets, 1998, Section 4.C]. Hence, as $(\bar{S}_c)$ are convex, so is their limit $S_\infty$ [Rockafellar and Wets, 1998, Prop 4.15]. It follows that $\phi_\infty$ is convex [Rockafellar and Wets, 1998, Ex 3.50].

Since $S_\infty$ is non-degenerate, there exists a radius $r_0$ such that $B(0, r_0) \subset S_\infty$, which implies that $\phi_\infty(\beta)$ is finite-valued for every $\beta$.

To prove point (ii), consider the sequence of functions $(\eta_c)_{c>0}$ formed by the indicators of convex sets $\bar{S}_c$:

$$\eta_c(\beta) = I_{\bar{S}_c}(\beta) = \begin{cases} 0 & \text{if } \beta \in \bar{S}_c, \\ +\infty & \text{otherwise.} \end{cases}$$

Note that the epigraph of $\eta_c$ is $\bar{S}_c \times \mathbb{R}_+$; these sets also converge to $S_\infty \times \mathbb{R}_+$ [Rockafellar and Wets, 1998, Ex 4.29], from which we conclude that function $\eta_c$ converge *epigraphically* to the indicator function $\eta_\infty$ of $S_\infty$ ($\eta_\infty = I_{S_\infty}$). We can then apply Attouch's theorem [Attouch and Beer, 1993, Combari and Thibault, 1998] ensuring that the graph of the subdifferentials of $\eta_c$

$$\mathcal{G}(\partial\eta_c) = \{(\beta, g) \,:\, g \in \partial\eta_c(\beta)\}$$

converge in Painlevé-Kuratowski sense to the graph $\mathcal{G}(\partial\eta_\infty)$ of subdifferential of $\eta_\infty$. This means that if a sequence $(\beta_c, g_c)_{c>0}$ such that $(\beta_c, g_c) \in \mathcal{G}(\partial\eta_c)$ for every $c > 0$ converges, then its limit belongs to $\mathcal{G}(\partial\eta_\infty)$.

Consider now a sequence $(\beta_s)_{s>0}$ satisfying the conditions described in (ii). Since it diverges to infinity and $\phi$ is coercive, we have $\phi(\beta_s) \to \infty$, and we may assume w.l.o.g that $\phi(\beta_s) > 0$ for all $s$.

We have by definition of sublevel sets that $\beta_s \in S_{\phi(\beta_s)}$, and therefore the gradient $\nabla\phi(\beta_s)$ belongs to the normal cone of $S_{\phi(\beta_s)}$ at $\beta_s$ (indeed, the gradient is orthogonal to the level sets; see e.g., [Courant and John, 1989, Chapter 1.5]). Since the normal cone of a convex set is the subdifferential of its indicator [Rockafellar, 1970, Section 23], we thus have

$$\nabla\phi(\beta_s) \in \partial I_{S_{\phi(\beta_s)}}(\beta_s), \tag{8}$$

Consider now the normalized levels sets as defined in (4). Denoting

$$\bar{\beta}_s = \frac{\beta_s}{R_{\phi(\beta_s)}},$$

we have $\bar{\beta}_s \in \bar{S}_{\phi(\beta_s)}$ and thus by simple rescaling (8) becomes

$$\nabla\phi(\beta_s) \in \partial I_{\bar{S}_{\phi(\beta_s)}}\left(\bar{\beta}_s\right).$$

Since $\partial I_{\bar{S}_c}$ is a cone (the normal cone to $\bar{S}_c$), this also holds for any positive multiple of $\nabla\phi(\beta_s)$. We deduce that for every $s > 0$

$$\left(\bar{\beta}_s, \frac{\nabla\phi(\beta_s)}{\|\nabla\phi(\beta_s)\|}\right) \in \mathcal{G}(\partial\eta_{\phi(\beta_s)}).$$

Note that since $\bar{\beta}_s$ belongs to the normalized level sets, this sequence is bounded. We can extract a subsequence $(\bar{\beta}_{s_k}, \frac{\nabla\phi(\bar{\beta}_{s_k})}{\|\nabla\phi(\bar{\beta}_{s_k})\|})_{k \geq 0}$ which converges to a limit point $(\hat{\beta}, \hat{g})$. By the previous remark on graphical convergence of subdifferentials, we have $(\hat{\beta}, \hat{g}) \in \mathcal{G}(\partial I_{S_\infty})$, i.e.,

$$\hat{g} \in \partial I_{S_\infty}(\hat{\beta}). \tag{9}$$

We need to prove that $\hat{\beta}$ is not 0. Since $\phi$ is strictly convex, the level set $\{\phi(\beta) = c\}$ is exactly the boundary of the sublevel set $\{\phi(\beta) \leq c\}$. Therefore, $\beta_s$ lies on the boundary of $S_{\phi(\beta_s)}$, and hence so does $\bar{\beta}_s$ lie on the boundary of $\bar{S}_{\phi(\beta_s)}$. Since 0 is in the interior of $S_\infty$, it also belongs to the interior of $\bar{S}_{\phi(\beta_s)}$ for $s$ larger than some $s_0$. Then, there exists $r_0 > 0$ such that $B(0, r_0) \subset \bar{S}_{\phi(\beta_s)}$ for $s \geq s_0$. By definition of boundary, we then have for $s \geq s_0$ $\|\bar{\beta}_s\| > r_0$, which leads to $\|\hat{\beta}\| > 0$.

To achieve the desired result; we need to relate $(\hat{\beta}, \hat{g})$ to $(\bar{\beta}, \bar{g})$. First, notice that by construction we have necessarily $\hat{g} = \bar{g} = \lim_{s \to \infty} \nabla\phi(\beta_s)/\|\nabla\phi(\beta_s)\|$. Then, note that

$$\bar{\beta} = \lim_{k \to \infty} \frac{\beta_{s_k}}{\|\beta_{s_k}\|}, \quad \hat{\beta} = \lim_{k \to \infty} \frac{\beta_{s_k}}{R_{\phi(\beta_{s_k})}}.$$

Taking the norm of the second limit, we have $\|\hat{\beta}\| = \lim_{k \to \infty} \frac{\|\beta_{s_k}\|}{R_{\phi(\beta_{s_k})}}$. Injecting back in the first limit yields

$$\bar{\beta} = \lim_{k \to \infty} \frac{\beta_{s_k}}{R_{\phi(\beta_{s_k})}} \cdot \frac{R_{\phi(\beta_{s_k})}}{\|\beta_{s_k}\|} = \frac{\hat{\beta}}{\|\hat{\beta}\|}.$$

Therefore, (9) becomes

$$\bar{g} \in \partial I_{S_\infty}(\|\hat{\beta}\|\bar{\beta}).$$

This means that $\bar{g}$ belongs to the *normal cone* of $S_\infty$ at $\|\hat{\beta}\|\bar{\beta}$ [Rockafellar, 1970, Sec. 23]. We note the level set $\{\beta : \phi_\infty(\beta) \leq \phi_\infty\left(\|\hat{\beta}\|\bar{\beta}\right)\}$ is exactly $\tau S_\infty$ for some $\tau > 0$. We use Corollary 23.7.1 from Rockafellar [1970] which states that if a vector is in the normal cone of the level set of $\phi_\infty$, then it must be a positive multiple of a subgradient. This implies that that there exists $\lambda \geq 0$ such that

$$\bar{g} \in \lambda\partial\phi_\infty(\|\hat{\beta}\|\bar{\beta}).$$

Finally, $\lambda > 0$ since $\|\bar{g}\| = 1$, and $\partial\phi_\infty(\|\hat{\beta}\|\bar{\beta}) = \partial\phi_\infty(\bar{\beta})$ by positive homogenity of $\phi_\infty$. $\qquad\square$

We can now prove our main result, which we restate here.

**Theorem 2.** *Assume that $\phi$ admits a non-degenerate horizon shape and let $\phi_\infty$ be its horizon function. Assuming that the following $\phi_\infty$-max-margin problem has a unique minimiser, then the mirror flow normalised iterates $\bar{\beta}_t = \frac{\beta_t}{\|\beta_t\|}$ converge towards a vector $\bar{\beta}_\infty$ and*

$$\bar{\beta}_\infty \propto \arg\min_{\bar{\beta} \in \mathbb{R}^d} \phi_\infty(\bar{\beta}) \quad \text{under the constraint} \quad \min_{i \in [n]} y_i \langle x_i, \bar{\beta} \rangle \geq 1,$$

*where the symbol $\propto$ denotes positive proportionality.*

The proof essentially follows exactly the same lines as in Section 3 but taking into account the fact that the loss is not exactly the exponential one.

*Proof.* Recall the definitions of the quantities $a_t$ and $q_t$ given above Equation (7) which enable to write:

$$\nabla\phi(\beta_t) = \nabla\phi(\beta_0) + Z^\top \int_0^t a_s q_s \mathrm{d}s.$$

Similar to the time change we performed in Section 3, we consider $\theta(t) = \int_0^t a_s \mathrm{d}s$. From Lemma 4, $\theta$ is a bijection over $\mathbb{R}_{\geq 0}$ and perform the time change $\tilde{\beta}_t = \beta_{\theta^{-1}(t)}$. Due to the chain rule, after the time change and dropping the tilde notation we obtain:

$$\nabla\phi(\beta_t) = \nabla\phi(\beta_0) + Z^\top \int_0^t q_s \mathrm{d}s.$$

Dividing by $t$ we get:

$$\frac{1}{t}\nabla\phi(\beta_t) = \frac{1}{t}\nabla\phi(\beta_0) + Z^\top \bar{q}_t, \tag{10}$$

where $\bar{q}_t := \frac{1}{t}\int_0^t q_s \mathrm{d}s$ corresponds to the average of $(q_s)_{s \leq t}$.

**Extracting a convergent subsequence:** We now consider the normalised iterates $\bar{\beta}_t = \frac{\beta_t}{\|\beta_t\|}$ and up to an extraction we get that $\bar{\beta}_t \to \bar{\beta}_\infty$. Since $q_t$ is a bounded function, up to a second extraction, we have that $q_t \to q_\infty$, and the same holds for its average: $\bar{q}_t \to q_\infty$. Taking the limit in Equation (10) we immediately obtain that:

$$\lim_t \frac{1}{t}\nabla\phi(\beta_t) = Z^\top q_\infty,$$

which also means that

$$\frac{\nabla\phi(\beta_t)}{\|\nabla\phi(\beta_t)\|} \xrightarrow[t\to\infty]{} \frac{Z^\top q_\infty}{\|Z^\top \bar{q}_\infty\|}$$

We can now directly apply Corollary 2 and there exists $\lambda > 0$ such that:

$$Z^\top q_\infty \in \lambda \partial \phi_\infty(\bar{\beta}_\infty)$$

The end of the proof is then as explained in Section 3.

$\square$

Finally we recall and prove Theorem 3 which provides a simple formula for the horizon function in the case of separable potentials.

**Theorem 3.** *Under Assumption 4, the potential $\phi$ admits a non-degenerate horizon shape and its horizon function is such that there exists $\lambda > 0$ such that for every $\bar{\beta} \in \mathbb{R}^d$:*

$$\phi_\infty(\bar{\beta}) = \lambda \lim_{\eta \to 0} \eta \cdot \varphi^{-1}\left(\phi\left(\frac{\bar{\beta}}{\eta}\right)\right).$$

*Proof.* **Lipschitzness, upper and lower boundedness.** For $\eta > 0$, let us denote by $h_\eta : \beta \mapsto \eta \cdot \varphi^{-1}(\phi(\beta/\eta))$ and notice that $\nabla h_\eta(\beta) = (\frac{\varphi'(\beta_k/\eta)}{\varphi'(\varphi^{-1}(\sum_i \varphi(\beta_i/\eta)))})_{k\in[d]} \geq 0$ Since $\varphi \geq 0$ and that $\varphi^{-1}$ and $\varphi'$ are increasing we get that that $\nabla h_\eta(\beta) \in [0,1]^d$. Therefore $(h_\eta)_{\eta>0}$ are uniformly Lipschitz-continuous. Consequently, for all $\beta$, $h_\eta(\beta)$ is upper-bounded independently of $\eta$. Lastly, since $\varphi \geq 0$, notice that $h_\eta(\beta) \geq \min_i |\beta_i| > 0$ for all $\beta \neq 0$.

**Point-wise and epi-convergence of $h_\eta$.** For all $\bar\beta$, by composition, $\eta \mapsto \eta \cdot \varphi^{-1}(\phi(\bar\beta/\eta))$ is a definable function, the monotonicity Lemma [Van den Dries and Miller, 1996] (Theorem 4.1) ensures that it has a unique limit in $\mathbb{R}$ which we denote $h_0(\beta)$. From the uniform Lipschitzness of $h_\eta$, we get that $(\eta, \beta) \in \mathbb{R}_{\geq 0} \times \mathbb{R}^d \mapsto h_\eta(\beta)$ is continuous. Hence for all sequence $\eta_k \to 0$, we get that $h_{\eta_k}$ epi-converges to $h_0$. Therefore $(\text{epi } h_{\eta_k})_k$ converges in the Painlevé–Kuratowski sense towards $\text{epi } h_{\eta_0}$.

**Link between the level sets of $h_\eta$ and those of $\phi$.** To conclude the proof it remains to notice that for all $c \geq 0$:

$$\{\beta \in \mathbb{R}^d, \phi(\beta) \leq c\} = \frac{1}{\eta}\{\bar\beta \in \mathbb{R}^d, h_\eta(\bar\beta) \leq \eta\varphi^{-1}(c)\}.$$

Therefore letting $\eta_c = 1/\varphi^{-1}(c)$ we get that

$$\eta_c \cdot S_c = \{\bar\beta \in \mathbb{R}^d, h_{\eta_c}(\bar\beta) \leq 1\}.$$

This simply means that $\eta_c$ is an appropriate normalising quantity, it replaces the normalisation by the radius of $S_c$. Since $\{\bar\beta \in \mathbb{R}^d, h_{\eta_c}(\bar\beta) \leq 1\}$ converges in the Painlevé–Kuratowski sense towards $\{\bar\beta \in \mathbb{R}^d, h_0(\bar\beta) \leq 1\}$, we get that $R_c\eta_c \cdot \bar{S}_c$ converges towards the same set. However, with our previous construction, we also have that $\bar{S}_c$ converges towards $\bar{S}_\infty$. The sets $\bar{S}_\infty$ and $\{\bar\beta \in \mathbb{R}^d, h_0(\bar\beta) \leq 1\}$ are therefore proportional and $h_0 \propto \phi_\infty$ which concludes the proof.

$\square$

