# OpenReview forum: "Implicit Bias of Mirror Flow on Separable Data"
_NeurIPS.cc/2024/Conference — NeurIPS 2024 poster_

### Official Review · Reviewer_Jqcr · 2024-07-05

**Soundness:** 3
**Presentation:** 3
**Contribution:** 3
**Rating:** 6
**Confidence:** 4

**Summary:**

In this paper, authors study the implicit bias of the mirror descent algorithm, from the perspective of the optimization trajectory of the continuous flow version. They propose the conceptions of horizon shape and horizon function $\phi_\infty$ to help characterize the properties of mirror flow at infinity. Since $\phi_\infty$ defines a norm, they prove that the mirror flow will eventually converge in direction to a max $\phi_\infty$-margin solution under the linear exponential-tail classification setting. Their findings contribute to a deeper understanding of the inherent characteristics of mirror descent across a wide range of potential functions.

**Strengths:**

1. The result of this work is solid, containing the general class of potential functions, and the authors derive a calculation rule of $\phi_\infty$ for a general class of potential functions.
2. The paper is informative and well-structured, particularly in section 3. By using the example of gradient descent within the framework established by the preceding lemmas, which is a special case of mirror descent, the authors clearly outline the reasons why mirror flow will eventually converge in direction without complicated formulas.

**Weaknesses:**

1. Since mirror descent is not so popular in the practice of machine learning problems,  there could be more discussion about the implications of their results. For example, Figure 1 is really interesting as it reveals that the mirror descent shares the same structure of implicit bias with the steepest descent [1], what is the essence of such similarities?
2. The setting of an infinitely small learning rate, i.e., optimization flow,  might be a little strong under a simple linear classification problem compared to the previous works. I suggest the authors state the technical challenges of the discrete optimization process of mirror descent.
3. I might be wrong, but it seems not strict to apply the Bolzano–Weierstrass theorem to an uncountably infinite set at page 5, line 160 and 61.

[1] Suriya Gunasekar, Jason Lee, Daniel Soudry, and Nathan Srebro. Characterizing implicit bias in terms of optimization geometry. In International Conference on Machine Learning, pages 1832–1841. PMLR, 2018.

**Questions:**

1. I'm really curious about whether this result could be extended to the q-homogenous neural networks like [1], cause it seems this result was also derived from the KKT condition of the lagrangian function.
2. I guess there might be no minus in line 461 for the first equality.
3. Could the author explain why they need to prove the uniqueness of the flow solution in Lemma 2, since this was not covered in the previous work, like [1][2]. Moreover, why do the authors need to prove that $\int_0^t a(\beta_s)ds \to \infty$ for Lemma 1.
4. What is the definition of $h_\infty$ in line 535 and $h(\cdot)$? Moreover, could the author explain more about how to derive formula (8) ?

[1] Kaifeng Lyu and Jian Li. Gradient descent maximizes the margin of homogeneous neural networks. In International Conference on Learning Representations, 2020.

[2] B.Wang, Q. Meng,W. Chen, and T.-Y. Liu. The implicit bias for adaptive optimization algorithms on homogeneous
neural networks. In International Conference on Machine Learning, pages 10849–10858. PMLR, 2021.

**Limitations:**

No limitations.

---

> ### Author Rebuttal · Authors · 2024-08-06
>
> We thank the reviewer for their thorough report and relevant comments. We will take them into account and make the appropriate changes in the revised version. Please find below our response to the comments.
>
> **Weaknesses**
>
> **W1:** Indeed, our result has connections with Theorem 5 from Gunasekar et al. (2018). Note first that the updates of steepest descent and mirror descent are different and one cannot be seen as a special case of the other (unless the geometry is Euclidean). For a norm $\Vert \cdot \Vert$, the steepest descent update is given by $w\_{t+1} = \text{arg min}\_{w} \ L(w\_t) + \langle \nabla L(w\_t), w - w_t \rangle + \Vert w -  w_t \Vert^2$  whereas the mirror descent update is given by $w\_{t+1} = \text{arg min}\_{w} \  L(w\_t) + \langle \nabla L(w\_t), w - w_t \rangle + D\_\phi(w, w_t)$  for a potential $\phi$.
>
> We can however observe the following connection: if the horizon function $\phi_\infty$ is a norm (this is the case when $\phi$ is symmetric), then **the implicit bias induced by mirror descent is the same as that of steepest descent with norm $\phi_\infty$**. In other words, $\phi$-mirror descent is asymptotically equivalent to $\phi_\infty$-steepest descent.
>
> Regarding the possible implications: our result could be used to study the mirror flow with new potentials that arise from gradient flow more complex neural network architectures, some of which are still under investigation (see answer to Q1 below).
>
> **W2:** The same results could be derived for mirror descent with finite step size, without major additional difficulties. We chose to study mirror flow as it simplifies the proofs and the presentation.
>
> **W3:** Indeed, we were (intentionally) a bit sloppy with the math, but the reasoning is correct. To make the derivation rigorous, one has to first consider any countable sequence $(t\_n)\_{n \in \mathbb{N}}$ with $t\_n \to \infty$ as $n \to \infty$ and then consider the sequence $(\bar{\beta}\_{t\_n}, q(\beta\_{t_n}))\_{n \in \mathbb{N}}$. We can then rightfully apply the Bolzano–Weierstrass theorem. Since the final limit $(\bar{\beta}\_\infty, q\_\infty)$ does not depend on the sequence $t_n,$ we can indeed conclude that $(\bar{\beta}\_{t}, q(\beta\_{t}))_{t \in \mathbb{R}}$ must converge towards it. We will clarify this argument in the revised version.
>
> **Questions**
>
> **Q1:** The link between gradient flow on $q$-homogeneous networks (as in [1]) and mirror flow (as in our setting) is still opaque. As discussed in our conclusion section, gradient flow on a depth $q$ **diagonal** linear network can be seen through the lens of [1], as well as through the lens of mirror flow (as in our paper). However, whether a link exists in more general settings is still unclear and remains a very interesting question for future work.
>
> **Q2:** It is indeed a typo, thank you for pointing it out.
>
> **Q3:** The main purpose of Lemma 2 is to ensure that the mirror flow solution exists and that it is well-defined for all $t \geq 0$. This fact is not trivial since, in a more general setting, the solution could "blow up in finite time". Previous work does not always prove global existence and implicitly assumes it to be true. Uniqueness is obtained as a by-product.
>
> Regarding the fact that $\int_0^t a(\beta_s)ds \rightarrow \infty$, it is an essential result for the main proof. We need it to apply the time change followed by the Cesaro argument (see lines 580 to 587 in the appendix).
>
> **Q4:** We made a notation error throughout the proof of Corollary 2: we used $h$ instead of $\phi$. We will fix this.
>
> Formula (8) stems from the fact that the gradient of $\phi$ is orthogonal to its level sets (we can invoke a result from multivariate calculus such as [1]). This means that $\nabla \phi(\beta_s)$ belongs to the normal cone of $S_{\phi(\beta_s)}$ at $\beta_s$. The subdifferential of the indicator of a convex set is given by its normal cone [2, Chap. 23], which allows to conclude. We will add these details to the revision.
>
> [1] Richard Courant; Fritz John (1999). Introduction to Calculus and Analysis Volume II/2.
>
> [2] R. Tyrell Rockafellar (1973). Convex Analysis.

---

> > ### Comment · Reviewer_Jqcr · 2024-08-09
> >
> > Thanks for the answers from the authors. I'm happy to maintain my score, voting for acceptance.

---

### Official Review · Reviewer_j6HU · 2024-07-08

**Soundness:** 3
**Presentation:** 3
**Contribution:** 3
**Rating:** 5
**Confidence:** 3

**Summary:**

This paper examines the implicit bias of mirror flow (the continuous-time counterpart of mirror descent) in the context of binary classification for linearly separable data. Given that the problem has infinitely many solutions, obtained at infinity, the authors aim to identify which solution is achieved through mirror flow. Assuming an exponential tail on the loss function, the authors demonstrate that mirror flow converges directionally to a maximum margin classifier, where the margin is characterized by a horizon function of the mirror potential. This result extends many existing works, and numerical experiments are provided to verify the findings.

**Strengths:**

1. The paper is well-written and well-organized. Despite the technical nature of the analysis and results, the paper is relatively easy to follow.
2. The paper is also well-motivated. Although mirror descent is not commonly used as an algorithm for training neural networks, analyzing its convergence is valuable for understanding the implicit bias of gradient descent in various neural network architectures.
3. I did not verify the details of the proof. However, the paper provides several motivating examples, including the quadratic potential corresponding to gradient flow, which makes the results quite convincing.
4. The main results extend several prior works.

**Weaknesses:**

Although the authors have stated that the convergence rate is left for future study, it would be beneficial to provide at least empirical evidence of the convergence rate. The authors mentioned in line 294 that the convergence rate varies across different potentials.

**Questions:**

1. In the main result, Theorem 2, the conclusion that the normalized mirror flow converges to a vector \(\bar{\beta}_{\infty}\) is drawn even when the \(\phi_{\infty}\)-max-margin problem does not necessarily have a unique solution. Can the authors provide more insight on this? If I understand correctly, in the motivating example, \(\bar{\beta}_{\infty}\) is a subsequence limit. If the \(\phi_{\infty}\)-max-margin problem does not have a unique solution, this result cannot be extended to the whole limit, unlike in the gradient flow case with quadratic potential.
2. More explanation could be provided for Figure 1. In particular, it would be interesting to see the trajectories of the mirror flows on the plane, rather than only showing the limit.

**Limitations:**

Yes

---

> ### Author Rebuttal · Authors · 2024-08-06
>
> We thank the reviewer for their report and relevant comments.
>
> **Weaknesses**
>
> The comment we make line 294 concerns the **training loss** convergence rate. We provide empirical evidence Figure 1 (left), we observe that the exact rate indeed depends on the potential. Also note that standard results from convex optimisation enables to show an $\tilde{O}(1 / t)$ upperbound on the rate (see proof of Proposition 1).
>
> Lines 303-304, we refer to the convergence rate of **the normalised iterates** towards the maximum margin solution, which we leave for future work. It would depend on the speed at which the normalised sublevel sets $\bar{S}\_c$ converge towards the limiting shape $S_\infty$.
>
> **Questions**
>
> **Q1:** We indeed need to assume that the $\phi_\infty$-max-margin problem has a unique solution, otherwise there could exist multiple subsequential limits. This is stated in the assumptions of Theorem 2. We also discuss sufficient conditions for this assumption to hold on lines 239-243.
>
> **Q2:** We will add more details concerning Figure 1 in Section 5. We agree with the reviewer that plotting the full trajectory is interesting and we will add it in the revised version.

---

> > ### Comment · Reviewer_j6HU · 2024-08-08
> >
> > I thank the authors for the response. My rating remains.

---

### Official Review · Reviewer_hMTV · 2024-07-11

**Soundness:** 4
**Presentation:** 4
**Contribution:** 4
**Rating:** 7
**Confidence:** 4

**Summary:**

This manuscript examines the implicit bias of mirror descent on a classification problem when the dataset is linearly separable. Assuming a coercive gradient, it demonstrates that the implicit bias is characterized by the shape of the level set of the mirror potential near infinity. Their analysis successfully recovers existing results for p-norm potentials and identifies the implicit bias of the potentials emerging in the analysis of linear neural networks. Additionally, it leaves the characterization of the implicit bias when the gradient is not coercive as an interesting open problem.

**Strengths:**

I think the paper is very well-written and has a solid contribution.

It addresses an important problem, aiming to understand the implicit bias of neural networks. Prior work has shown that the dynamics of linear networks can be characterized by mirror descent, highlighting the relevance of this study.

**Weaknesses:**

NA

**Questions:**

In line 123, the authors suggested that the logistic loss satisfies the conditions in Assumption 1. However, it is clear that the logistic loss does not have an exponential tail. Could they clarify whether this is a mistake or if there is an underlying argument supporting their claim?

---

> ### Author Rebuttal · Authors · 2024-08-06
>
> We thank the reviewer for their positive report and relevant comments.
>
> It can be verified that the logistic loss indeed satisfies the exponential-tail condition of Assumption 1, since $\ln(1+\exp(-z))$ is equivalent to $\exp(-z)$ as $z \rightarrow + \infty$.

---

> > ### Comment · Reviewer_hMTV · 2024-08-11
> >
> > I thank the authors for the response. My rating remains.

---

### Official Review · Reviewer_DbGr · 2024-07-16

**Soundness:** 4
**Presentation:** 4
**Contribution:** 3
**Rating:** 7
**Confidence:** 2

**Summary:**

This paper considers the asymptotic behaviour of the mirror descent (MD) algorithm for a linear classification task. It is shown that the classifier (hyperplane orthogonal to $\beta$) will be a max-margin classifer, where the margin is determined by some unknown horizon function $\phi_\infty$. This works extend prior work which consider $\ell_p$ and homogeneous potential functions for MD, and shows this result for very general $\phi$.

**Strengths:**

The paper makes an interesting statement about behaviour of mirror descent on classification tasks, will minimal assumptions. In doing so, it takes a big step and extends previous work to the cover general potential functions.
The paper is well written and the figures help with understanding the concepts of convergence.

---
*While I could understand the paper, this is not my area of research. I do not find myself fit to evaluate the paper on soundness, relevance to the sub-field and importance of contributions.*

**Weaknesses:**

- The paper does not characterize $\phi_\infty$ in terms of the bregman potential $\phi$ (and other relevant entities).
The main result expresses that there exists some function, that is minimized by $\bar \beta_\infty$, the direction of the classifier as $t\rightarrow \infty$.
I think this limits the relevance and strength of the result. For instance, this does not help with interpretability compared to the case where we can prove the optimization algorithm converging to a max-margin classifier wrt the $\ell_2$ norm.

- I am not sure about relevance and use-cases of the mirror descent algorithm with very general potentials. As far as I know, typically, a small set of norm-based or entropic (neg-ent, tsallis, etc) are used within applications of ML. So while the theorem makes an interesting statement for an optimization standpoint, I'm not sure how relevant it is for the ML community. The theorem is also not entirely relevant to the pure optimization community since it's for the specific case of linear classification with finite data.
---
*While I could understand the paper, this is not my area of research. I do not find myself fit to evaluate the paper on soundness, relevance to the sub-field and importance of contributions.*

**Questions:**

- Are there any classes of potential functions (other than the norms and $L$-homogeneous ones), for which $\phi_\infty$ may be calculated or approximated?

- Beyond gradient descent, is there any work that quantifies the rates of convergence to max-margin classifiers? Is this even possible?

**Limitations:**

The limitations are discussed.

---

> ### Author Rebuttal · Authors · 2024-08-06
>
> We thank the reviewer for their report and relevant comments.
>
> **Q1:** In Theorem 3, we provide a formula for computing the horizon function when the potential $\phi$ is **separable**. In that case, $\phi_\infty$ can be obtained by computing the limit at infinity of a simple one-dimensional function. We apply this to two examples of non-homogenous potentials in Section 5: the hyperbolic entropy and the cosine entropy.
>
> For non-separable potentials, there is a priori no straightforward formula and one has to find the limiting shape $S_\infty$ towards which the normalised sublevel sets $\bar{S}_c$ converge to as $c \to \infty$. However, we emphasize that the separable case covers almost all known examples.
>
> **Q2:** By building on the arguments of Proposition 1 (line 467), we could show that $L(\beta_t)$ decreases with rate $\tilde{O}(1/t)$. However, this convergence rate does not inform us on the rate of convergence of the normalized iterates $\bar \beta_t$ to the limiting direction. This question is indeed a challenging open problem and is left for future work.

---

> > ### Comment · Reviewer_DbGr · 2024-08-08
> > **Reviewer Response**
> >
> > Thank you for the answers.
> >
> > Base on this, and reading other reviews, I have updated my review (the confidence score).

---

### Comment · Area_Chair_QgKv · 2024-08-07

Dear Reviewers,

Now that the rebuttal period has ended, please take a moment to review the authors' responses to your initial reviews. Your engagement is crucial for:

    Ensuring a fair evaluation process
    Improving the quality of final reviews
    Fostering meaningful scientific discourse

If you have any questions or need clarification, please post follow-up comments.

Your continued dedication to this process is greatly appreciated.

Best regards,

AC

---

### Decision · Program_Chairs · 2024-09-25

**Decision:**

Accept (poster)

**Comment:**

This paper examines the implicit bias of mirror flow (continuous-time mirror descent) for linearly separable classification problems. The main result shows that, under mild assumptions, mirror flow converges in direction to a maximum margin classifier characterized by the horizon function of the mirror potential. This extends previous work to more general potential functions.

Strengths:
- Solid theoretical contribution extending prior results
- Well-written and structured paper
- Relevant to understanding implicit biases in optimization

Weaknesses:
- Limited discussion of practical implications
- Some technical details need clarification

The reviewers were generally positive, with ratings ranging from 5 to 7. They found the paper to be technically sound with good to excellent contributions. However, several points require attention before publication:

1. Expand discussion on practical implications and connections to related work
2. Clarify some technical details in proofs and notations
3. Add more explanations for figures, especially trajectories in Figure 1
4. Fix minor typos and notation errors identified by reviewers

Overall, this paper makes a valuable contribution and is recommended for acceptance, contingent on thorough revisions addressing the above points. The authors should carefully consider all reviewer comments when preparing the final version.